# Development and Optimization of a High Sensitivity LC-MS/MS Method for the Determination of Hesperidin and Naringenin in Rat Plasma: Pharmacokinetic Approach

**DOI:** 10.3390/molecules25184241

**Published:** 2020-09-16

**Authors:** Jesús Alfredo Araujo-León, Rolffy Ortiz-Andrade, Rivelino Armando Vera-Sánchez, Julio Enrique Oney-Montalvo, Tania Isolina Coral-Martínez, Zulema Cantillo-Ciau

**Affiliations:** 1Chromatography Laboratory, Chemistry Faculty, Autonomous University of Yucatan, Mérida 97069, Mexico; rivelaino02@hotmail.com (R.A.V.-S.); oney27@gmail.com (J.E.O.-M.); tcoral@correo.uady.mx (T.I.C.-M.); 2Pharmacology Laboratory, Chemistry Faculty, Autonomous University of Yucatan, Mérida 97069, Mexico; rolffy@correo.uady.mx; 3Medicinal Chemistry Laboratory, Chemistry Faculty, Autonomous University of Yucatan, Mérida 97069, Mexico; zulema.cantillo@correo.uady.mx

**Keywords:** ESI, flavonoid, optimization, pharmacokinetic, validation, QqQ

## Abstract

The purpose of this study was to develop, optimize, and fully validate a high-sensitivity methodology using UHPLC-MS/MS to simultaneously quantify hesperidin and naringenin in microsamples (100 µL) of murine plasma after intragastric administration of single pure flavonoids and a mixture. The optimization process allowed for high sensitivity with detection limits of approximately picogram order using an electrospray ionization (ESI) source in negative mode and an experiment based on multiple reaction monitoring (MRM). The validation parameters showed excellent linearity and detection limits, with a precision of less than 8% and a recovery of over 90%. This methodology was applied to compare the pharmacokinetic parameters for the administration of hesperidin and naringenin in individual form or in the form of a mixture. The results showed an absence of significant effects (*p* > 0.05) for Tmax and Cmax; however, the AUC presented significant differences (*p* < 0.05) for both flavonoids when administered as a mixture, showing an improved absorption ratio for both flavonoids.

## 1. Introduction

Flavonoids are a selected group of polyphenolic compounds widely contained in citrus fruits; according with Tripoli and coworkers [1] cover over 60 types have been identified. The main classification of these compounds are as glycosides (neohesperidosides and rutinosides) and aglycones as naringenin and hesperetin [1]. A recent investigation demonstrated that naringenin increases locomotor activity and reduces diacylglycerol accumulation [2]; hesperidin affects the microbial spoilage and antioxidant quality [3]. A derivate from heptamethoxyflavone induces a brain-derived neurotrophic factor via cAMP by extracellular signal-regulated kinase [4]; other flavonoids have antiproliferative activities in cancer cell lines [5]; antinociceptive and anti-inflammatory effect [6], and activities in Alzheimer’s disease [7].

Recent research has focused on studying a mixture with two or more flavonoids to increase or improve the pharmacological effect. A hesperidin/naringenin mixture exhibits vasorelaxant and antihypertensive properties [8], antioxidant activity, and a membrane phospholipid composition [9], also inhibiting HER2 tyrosine kinase activity [10]. A mixture of apigenin, hesperidin, and kaempferol was demonstrated to reduce the lipid accumulation in human adipocytes [11]. A natural mixture from propolis has shown antioxidant, antiproliferative, antitumor, and anti-inflammatory activities [12].

Pharmacokinetic studies are currently an essential parameter for explaining the bioavailability and distribution of molecules with high pharmacological potential in crude extracts [13] or as isolated metabolites [14,15,16]. For all tests in preclinical studies, it is necessary to include these parameters. Generally, rodent evaluation is the most popular form of experimentation; however, in almost all cases, the sacrifice of the animal is necessary to obtain a representative sample of heart blood. Only a few studies have reported the use of small samples of between 30 and 200 µL of serum or plasma collected by capillary microsampling from the vein cannula or tail-vein [17,18,19].

High-performance liquid chromatography (HPLC) coupled with mass spectrometry (MS) is the primary technique that has been used for pharmacokinetic studies, especially tandem mass spectrometry (MS/MS), in complex matrices as biological fluids or total blood when high quality control is necessary for the quantification process with high sensitivity and selectivity. Previous studies on the pharmacokinetics of polymethoxylated flavones [20], phenolic acids, and flavonoids [21] used HPLC-MS/MS to quantify rat plasma analytes. The other application, the HPLC-MS/MS in pharmacokinetic studies with high selectivity, is in organic crude extracts [13,14,15] due to the fact that it is only necessary to quantify one or a few flavonoids of the total molecules of the extract. 

Mix-160 is a molar mixture of hesperidin and naringenin with an effective dose of 161 mg/kg, for which, in our working group, we observed significant vasorelaxant, antihypertensive, and antihyperglycemic effects [8]. As a part of a preclinical step, it is crucial to study the pharmacokinetic parameters of these flavonoids and compare the oral administration for the single pure flavonoids, and, in the mixture, to explain the bioavailability of the hesperidin/naringenin mixture, which has, to date, not been reported in the literature. For these reasons, the aim of our work is to develop a high sensitivity analytical methodology using HPLC-MS/MS with detection limits in the picogram-order to extract hesperidin and naringenin from rat plasma (microsample of 100 µL). Finally, we use the developed method to determine the pharmacokinetic parameters in this study.

## 2. Results and Discussion

### 2.1. Optimization of the Ionization Source

#### 2.1.1. Plackett–Burman Design (PBD)

As shown in the chart (Figure 1), PBD showed that the gas (N_2_) temperature and capillary voltage did not have a significant effect (*p* > 0.05). However, the other five parameters did have a significant effect (*p* < 0.05) over the ionization source. The factor with the most statistical relevance was the sheath gas flow (SGF). The SGF, gas flow and sheath gas temperature (SGT) are related to the UHPLC flow proportionality, in this case at 600 µL/min. Generally, for UHPLC-MS systems, the flow rates range between 200 µL/min to 500 µL/min; however, the ESI-jet stream allows for flow rates of approximately 1500 µL/min enabling excellent desolvation. The SGF and gas flow had a positive effect (+1) on the ionization; however, since MS operating conditions allow for only a maximum SGF and gas flow of 12 L/min and 13 L/min, respectively, these parameters could not be optimized any more.

The nozzle voltage was the second factor with more relevance; this parameter is the voltage applied with a differential dielectric potential over the electrospray syringe and with the skimmer over the electrode. In this case, the ionization charge for flavonoids was on the negative mode, with a specific adduct [M−1].

According to Szerkus and coworkers [22], small molecules with ionizable functional groups, such as dexmedetomidine, containing an aromatic structure very much like the one displayed by the flavonoids, enables a better ionization when the following parameters are met: gas temperature over 300 °C, nebulizer pressure over 40 psi, and capillary voltage ranging between 1500 V and 2000 V. An increased voltage reduces the peak area, which is directly related to the nozzle voltage.

Flow rate has an essential role in ionization; in this case, the flow was 600 µL/min, which was considered as an intermediate flow rate. The nebulizer gas (NG) and SGF are both related to the flow rate; when the flow rate increases, NG must be increased, while a higher NG with lower flow generated small peaks and a loss of sensitivity. In addition, the flow rate interacted with the drying gas temperature and the SGT; for this interaction, it was necessary to increase both parameters to obtain a better desolvation. To improve the ionization in ESI, the N_2_ gas must be supplied with enough pressure (40 psi) and with a high SGF [23].

#### 2.1.2. Box–Behnken Design (BBD)

The nozzle voltage (NV), NG and SGT were selected as critical factors and optimized in the Box–Behnken design; BBD is a second-order multifactorial design based on an incomplete 3^3^ factorial design. An estimation of the mathematical relationship between the response and the critical factors was possible using only 15 experiments. ANOVA and multiple regression analysis was performed to obtain the regression coefficients and the central equation for BBD: (1)$$Y=2398-328.12x_{1}+121.5x_{2}+325.37x_{3}+35.87x_{1}^{2}-45x_{1}x_{2}-52.75x_{1}x_{3}-23.38x_{2}^{2}+10.5x_{2}x_{3}-74.63x_{3}^{2}$$

The partial derivative equations shown below were used to obtain the critical points for the surface response: (2)$$\frac{\delta Y}{\delta x_{1}}=-328.12+71.74x_{1}-45x_{2}-52.75x_{3}$$
(3)$$\frac{\delta Y}{\delta x_{2}}=121.5-45x_{1}-46.76x_{2}+10.5x_{3}$$
(4)$$\frac{\delta Y}{\delta x_{3}}=325.37-52.75x_{1}+10.5x_{2}-149.26x_{3}$$

The calculated values for the critical points are: NV = 1500 V, Nb = 40 psi, and SGT = 400 °C; the ANOVA regression model demonstrates 97% ($R_{ajs}^{2}$) of security to explain the variations of the model with a satisfactory relationship between the experimental and predicted values for the response. Another parameter for the ANOVA was the lack-of-fit; this parameter indicates whether or not the model adjusts with a second-order equation, in this case, it did not show significant differences (*p* = 0.184), which confirms that all the data were adjusted to a second-order model.

A comparison of the central points of the PBD with the optimal points of the BBD showed more sensitivity for the chromatogram with an increased 1:15 ratio (Figure 2) for the flavonoids injected into the same concentration solution (10 ng/mL). The chromatogram shows that the results have the same effect in that they increase the area when the method is applied under optimal conditions with glycoside and aglycone structure flavonoids.

### 2.2. Validation Parameters

#### 2.2.1. Specificity

No significant interfering peaks were observed in representative chromatograms of Figure 2 when comparing the blank plasma versus fortified plasma spiked with the analytes. It was an excellent chromatographic resolution (>1.5) between hesperidin, naringenin, and the internal standard (quercetin) was observed. The mass spectrometry of triple quadrupole (QqQ) has higher specificity with multiple reaction monitoring (MRM) experiments to eliminate the matrix’s interferences.

#### 2.2.2. Linearity, Limited of Detection (LOD), and Limited of Quantification (LOQ)

For each chromatographic analysis, two microliters of standards solution or samples were injected on-column and calibration curves were constructed from the average areas for each peak, using the product ion in the MRM experiment. The slope (b) and intercept (a) were calculated using Student’s *t*-test, with a confidence interval, residual variance, lack-of-fit test applied using ANOVA; the correlation coefficient (r) exceeded 0.99 (Table 1). 

Statistical analysis and graphic inspection showed a proportional increase with excellent correlation. It also determined a nonzero slope value, where the intercept crosses through the origin, while the ANOVA, more specifically, the lack-of-fit test, did not exhibit any significant difference (*p* > 0.05). These results, compared to the FDA guides, confirmed the linearity for each flavonoid with a first-order linear model.

Recent studies have reported a LOD for six flavonoids in plants ranging from around 0.058–1.88 µg/mL [24]; however, these limits are not enough for pharmacokinetic studies where the sample is only a few microliters of serum. Bhatt and coworkers [25] reported a LOD for hesperidin of 60 ng/mL using MS/MS. A recent work for green and red propolis using an HPLC coupled with MS of QqQ reported a LOD of 0.4 ng/mL for naringenin, 0.66 ng/mL for rutin, and approximately 0.7 ng/mL to 15 ng/mL for other flavonoids, such as kaempferol, luteolin, and pinocembrin [26]. Another work based on flavonoid quantification in kiwi fruits using UPLC-QqQ-MS/MS reported a LOQ of 80 ng/mL and 81 ng/mL for rutin and quercetin, respectively [27]. Based on these reports, our methodology shows more sensitivity than recent works reported in the literature; we consider that this was made possible due to the optimization process used to increase the sensitivity.

A highly sensitive pharmacokinetic study of estrogens using a small microliter blood sample and UHPLC-MS/MS previously reported a LOD of 2.5 ng/mL [28]; Another study quantified epicatechin and procyanidin B2 in rat plasma after oral administration [14] and reported a lower LOQ of 5 ng/mL with a precision lower than 2% and a recovery higher than 90%. Finally, a pharmacokinetic study of rutin and quercetin in rats after oral administration has reported a LOQ 55.5 ng/mL [13]. However, the evaluation of the nominal concentration with precision and accuracy reported values ranging between 208 ng/mL and 4160 ng/mL for rutin and 191 ng/mL to 3820 ng/mL for quercetin. For determination of eriodicytol-6-C-β-D-glucoside (flavanone as hesperidin) a LOD of 0.11 ng/mL (2.75 pg) has been reported [15], and, finally, for myricetin and derivatives, a LOD of 10 ng/mL (approximately 50 pg to 200 pg has been reported) [16]. In comparison, our results showed that the LOD from each flavonoid evaluated in a matrix (murine plasma) presents a better LOD and LOQ in picogram order.

#### 2.2.3. Precision and Accuracy

The precision and accuracy data are shown in Table 2; these results were compared with the guidance from the FDA, ICH, and AOAC peer verified method, and, finally, Horwitz’s function was used to demonstrate that the developed method presents a high accuracy.

Samples of 100 µL of murine plasma were fortified with 2.5 ng/mL, 7.5 ng/mL, and 12.5 ng/mL, respectively; this evaluation for the precision was evaluated intraday (*n* = 9) and interday (*n* = 27) and reported as the relative standard deviation (RSD); the accuracy was declared as the mean of the recovery percent for intraday and interday evaluation. The international guidance reported in the literature at these concentrations shows a recovery range between 60% and 115% with a precision of 21%–32% [29,30]. According to the results (Table 2), the optimized method and analytical extraction by solid-phase extraction showed an excellent recovery of 86.58%–97.32% with a high precision of lower than 10%.

#### 2.2.4. Recovery and Matrix Effect

The peak area ratios measured the extraction recoveries in three quality control concentrations (5, 50, and 500 ng/mL). The results of recovery were all over 88.35% with RSD less than 10.6%. Matrix effect is defined as ion suppression or enhancement of matrix on analyte ionization. The results (Table 3) show that no endogenous substance in rat plasma has a significant effect; no interference was obtained in the ionization process on the quantification of hesperidin, quercetin (IS), and naringenin. The calculated values had observed ion enhancement for the three flavonoids; the matrix effect was 102.7 ± 2.02% for hesperidin, 104.7 ± 2.21% for quercetin (IS), and 103.0 ± 0.97% for naringenin.

### 2.3. Pharmacokinetic Assessment

According to the results, the LOD, precision, and accuracy were found to be favorable for the application of the pharmacokinetic assessment of hesperidin and naringenin in rat plasma after intragastric administration; the calculated parameters were a maximum time in plasma concentration (T_max_), maximum plasma concentration (C_max_), half-life time (T_1/2_), the plasma exposure (area under the curve (AUC0-24)), mean residence time (MRT), clearance (CL), and distribution volume (V), as shown in Table 4.

To compare the pharmacokinetic parameters for the administration of the single and mixed flavonoids, we used a T-test (*p* = 0.05) for each parameter. The T_1/2_ for hesperidin (Hsn) does not show differences (*p* < 0.05) in both administrations, which is the same case for naringenin (Ngn); however, we observed a reduction of 1.04 h when adminisering Ngn/Hsn, determining that the elimination is the fastest. As shown in the graphic (Figure 3), we calculated the slope to Hsn and Ngn in both administrations between Tmax to the final time; Hsn does not show differences, however, in Nsn evaluation, the slope (−54.5) for Hsn/Ngn (Figure 3B) is more pronounced than that in the single administration (−29.8). Since T_1/2_ is dependent in the V and CL, we observed (Table 4) that V decreases near to half of the original value (single administration) and CL increases to almost twice the ratio, evidencing an increased clearance realized by excretion or metabolism because V is less in Ngn/Hsn, with a higher concentration in plasma (C_max_). Although C_max_ does not show any differences (*p* < 0.05) between each administration for Hsn and Ngn, in both cases, we observed an increment in C_max_ when each flavonoid was administered in a mixture with a ratio of 1.78 and 1.52 for Hsn and Ngn, respectively.

The AUC_0–24_ comparison between the Hsn and Hsn/Ngn administrations showed a significant difference (*p* < 0.05); a similar case was observed in Ngn and Ngn/Hsn. The mixture administration of both flavonoids does not modify the T_max_; the C_max_ did not showed a significant difference (*p* > 0.05), but the concentration of hesperidin increases around 1.6 times when administered in the mixture, the same case was observed with naringenin; the rate of absorption is higher and better, with increased concentration in plasma for both flavonoids when the administration involves the mixture. The results showed that Ngn has a quicker and better absorption ratio than Hsn. The flavonoids show poor solubility in water, affecting the absorption ratio in the gastrointestinal tract; a study for the tissue distribution showed an abundant concentration in the gastrointestinal tract, jejunum, ileum, and colon, which furthermore reported that the distribution in tissue is high compared with plasma after oral administration [31].

The Hsn (glycoside) has a low-hydrolysis in the intestine, which delays the adsorption compared with the Ngn (aglycone), which is recovered in plasma mainly as glucuronides with a much faster adsorption ratio. These effects are responsible for the main differences observed in the pharmacokinetic parameters related to the absorption (AUC, C_max_, and T_max_). The distribution of these flavonoids, furthermore, has many pathways as tissue absorption in the trachea and lung. Finally, for the flavonoids, it has been reported that the absorption sites in the small intestine and colon can undergo alteration, decreasing the absorption ratio, especially for glycosides [31,32,33].

## 3. Materials and Methods 

### 3.1. Chemicals and Reagents 

The methanol (MeOH), acetonitrile (ACN) and water (J.T. Baker, Phillipsburg, NJ, USA) used in the chromatographic analysis were of chromatography grade; the formic acid, and dimethyl sulfoxide (DMSO, 99%; Sigma-Aldrich, St. Louis, MI, USA) used in the study was of analytical grade (ACS). Hesperidin, hesperetin, and quercetin were of analytical grade (≥98%, Sigma-Aldrich, St. Louis, MI, USA).

### 3.2. Animal Facility Conditions

Male Wistar rats (200–250 g) were obtained from the “Universidad Juárez Autónoma de Tabasco” bioterium, and kept in polycarbonate cages under standard laboratory conditions (12 h light and dark cycles, room temperature controlled at 25 °C and a humidity percentage of approximately 45 to 65%). Rodents were fed with a certified rodent diet and tap water ad libitum. Animal procedures were conducted in accordance to the Mexican Federal Regulations for Animal Experimentation and Care (NOM-062-ZOO-1999) and in accordance with the Institutional Animal Care and Use Committee guidelines as stated in the US National Institute of Health publication (no. 85-23, revised 1985). The experiments were approved by the Animal Ethics Committee of “Universidad Juárez Autónoma de Tabasco” university (Code:2017-001, Approved: October 2017).

### 3.3. Optimization of Ionization Source 

#### 3.3.1. Plackett-Burman Design

A Plackett-Burman design (PBD, 2^8 × 3/24) based on the first-order model was applied to identify the significant factors (*p* < 0.05) for the ionization source and increase the MS sensitivity for each flavonoid. Seven factors were selected for evaluation (Table 5), using −1, 0, and +1 as code variables. The individual effect of each factor was calculated as follows: (5)$$E\left(X_{i}\right)=2\;\left(\sum M_{i}^{+}-\sum M_{i}^{-}\right)/N$$
where $E\left(X_{i}\right)$ is the effect in the tested variable, $M_{i}^{+}$ and $M_{i}^{-}$ are responses to trials in which the variable is at its high or low level, respectively, and N is the total number of trials. The main effects were checked by a Pareto chart, and Statgraphics Centurion XV for Windows (The Plains, VA, USA) was employed for data analysis.

#### 3.3.2. Box–Behnken Design

To examine the PBD design and determine the significant factors (*p* < 0.05), the optimization of these factors was performed using a Box–Behnken design (BBD)—a second-order model for three factors (Table 6). The design consisted of 17 experiments, 12 from the BBD and five replicates at the central point of the statistical design, which was used to allow the estimation of the pure error sum of the squares. The dependent and independent variables were coded according to factorial design. The software Statgraphics Centurion XV for Windows was employed for experimental design, data analysis, and model building. The experiments were analyzed by multiple regression to fit into the following nonlinear quadratic polynomial model. This model contains the following terms: (6)$$Y=b_{0}+\overset{k}{\underset{i=1}{\sum }}b_{i}X_{i}+\overset{k}{\underset{i=1}{\sum }}b_{ii}X_{i}^{2}+\overset{k-1}{\underset{i=1}{\sum }}\overset{k}{\underset{j=i+1}{\sum }}b_{ij}X_{i}X_{j}+\varepsilon$$
where *Y* is the yield, *b_o_*, *b_i_*, *b_i_**_i_* and *b**_ij_* are the regression coefficients for the intercept, linear, quadratic and interaction terms, respectively, and *X_i_* and *X_j_* are the independent variables.

### 3.4. Validation

The method was validated following the Food and Drug Administration (FDA) guidelines given in the document “Analytical Procedures and Methods Validation for Drugs and Biologics” with respect to specificity, linearity range, limit of detection (LOD), limit of quantification (LOQ), precision, and accuracy.

#### 3.4.1. Linearity Range 

Linearity was studied using five different amounts of each flavonoid (within the range of 5–1000 ng/mL). A calibration curve was also generated using a linear regression of plot peak area versus the amount injected into the HPLC column.

#### 3.4.2. LOD and LOQ

The LOD and LOQ was calculated to create a new calibration curve (0.5–10 ng/mL) using the parameters shown in Equation (1):(7)$$L=\frac{k\left(S_{y,x}\right)}{b\sqrt{n}}$$

In Equation (7), *L* is the LOD or LOQ, *k* is a constant (i.e., LOD has a value of 3, LOQ has a value of 10), $S_{y,x}$ is the residual standard deviation, and *b* is the slope.

#### 3.4.3. Precision and Recovery

Recovery was calculated using the bias percentage. The peak areas for each flavonoid standard were compared with the matrix recovery. The precision was evaluated using the relative standard deviation (RSD). Evaluations were performed intraday (*n* = 9) and interday (*n* = 27). Each flavonoid was spiked at 2.5 ng/mL, 7.5 ng/mL, and 12.5 ng/mL into 100 µL of murine plasma, and the recovery was evaluated with a reference standard solution at the same concentration.

#### 3.4.4. Recovery and Matrix Effect

Three quality control levels (QCL) were evaluated comparing the peak areas (*n* = 6). For the recovery was calculated comparing the means of the analytes spiked before extraction (R3) between analytes spiked post-extraction (R2), in the three QCL. The acceptable relative standard deviation (RSD) of the peak area for each flavonoid should be ≤15%. The matrix effect was calculated comparing the means of analytes spiked post-extraction (R2) between analytes in the pure standard solution (R1), at the same QLC. The matrix effect values are considered ionization suppression if less than 85% and ionization enhancement if more than 115% [34].

### 3.5. Flavonoid Extraction from Murine Plasma 

Hesperidin, quercetin, and naringenin extraction was conducted by solid phase extraction (SPE) using SPE cartridges (Agilent Technologies, San Jose, CA, USA) filled with 1000 mg of C18 and 6 mL of capability. For the conditioning step, 5 mL of MeOH and 5 mL of water was used. For loading of the sample, murine plasma was dissolved in 5 mL of water, passed through to the cartridge, and then subsequently washed with 10 mL of water. Finally, the elution was carried out with a 2 mL solution of MeOH:DMSO (8:2, *v*/*v*). All experiments were conducted under controlled pressure in a Visiprep SPE vacuum distribution manifold (15 inHg at 1 mL/min).

### 3.6. HPLC-MS/MS Instrumentation

For the HPLC-MS/MS analysis, a quaternary pump (Agilent Technologies 1290-series, Agilent, San Jose, CA, USA) was coupled to a mass spectrometer (Agilent Technologies, 6470 model, San Jose, CA, USA) equipped with a jet stream electrospray ionization source (ESI source) operated in negative mode. For flavonoid detection, the optimal parameters for the QqQ mass spectrometer were set as follows: gas temperature at 350 °C; gas flow at 13 L/min; nebulizer to 40 psi; sheath gas temperature (SGT) at 400 °C; sheath gas flow (SGF) at 12 L/min; capillary voltage of 3000 V; and nozzle voltage of 1500 V. Spectra were recorded in negative-ion mode by conducting a multiple reaction monitoring experiment (MRM). Table 7, below, shows the transition ions and MS parameters for the MRM.

For flavonoid separation, two microliters of the solution was a load on the column; the LC-MS system was equipped with a C18 column (50 mm × 3.0 mm, internal diameter 1.8 µm; Zorbax Eclipse Plus C18 RRHD, Agilent, USA). Chromatography was performed under gradient conditions, where the water used contained 0.1% formic acid, with a mobile-phase flow rate of 0.6 mL/min. The gradient program started with 75% of water, 12.5% MeOH, and 12.5% ACN. At minute 7, mobile phase proportions were found at 63% of water, 18.5% MeOH, and 18.5% ACN. Finally, at minute 7.01, mobile-phase proportions were maintained at 50% MeOH and 50% ACN for 5 min more while the mobile phase continued until column cleanup. The column was purged with MeOH, followed by a 10 min equilibration lapse with the initial mobile phase proportions. 

### 3.7. Pharmacokinetic Studies and Data Analysis

Experimental groups were divided into hesperidin, naringenin, and a mixture (hesperidin/naringenin). Before oral administration, 51 male Wistar rats were randomly assigned in 17 groups (*n* = 3, each group corresponds to the sample collection time), and 12 h before initiating the study, the rats were restrained to avoid food consumption, but with water still available. The administration was carried out using intragastric gavage at single doses of hesperidin (69 mg/kg), naringenin (92 mg/kg), and the mixture (161 mg/kg). Blood microsamples (0.5 mL) were collected from tail-vain at 5 min, 15 min, 30 min, 60 min, 90 min, 120 min, 180 min, 240 min, 300 min, 360 min, 420 min, 480 min, 540 min, 600 min, 720 min, 1080 min, and 1440 min and then transferred to a 2 mL centrifuge tube. After the samples had coagulated they were centrifuged at 3000 rpm for 10 min; finally, the plasma were separated and stored at −80 °C until analysis. Pharmacokinetic parameters were calculated by noncompartmental analysis of plasma concentration versus time using WinNonlin 2.0 software (Certara USA, Inc., Princeton, NJ, USA). To compare pharmacokinetic parameters between single flavonoids and the mixture, a t-test was carried out with SPSS 20.0 (IBM Corp., Armonk, NY, USA) software (*p* = 0.05).

## 4. Conclusions

In this study, we developed an optimized and validated methodology with high sensitivity of picogram order, and demonstrated that via an optimization process (PBD and BBD) with response surface methodologies, it is possible to increase the signal ratio by more than 15 times the standard evaluation. The developed method was used to evaluate the main pharmacokinetic parameters, which showed that when hesperidin and naringenin were administered in a mixture, a better absorption ratio and, thus, an increased plasma concentration was realized as a function of time. The T_max_ and C_max_ do not present significant variations in both administrations. However, the C_max_ was found to increase around 1.6 times when the flavonoids were administered into the mixture. Finally, for this method, we considered a bioethical analytical process because the experiments were performed by tail venipuncture, avoiding rodent sacrifice in each assay due to the use of only 100 µL of murine plasma to determine the concentration of hesperidin and naringenin.

## Figures and Tables

**Figure 1 molecules-25-04241-f001:**
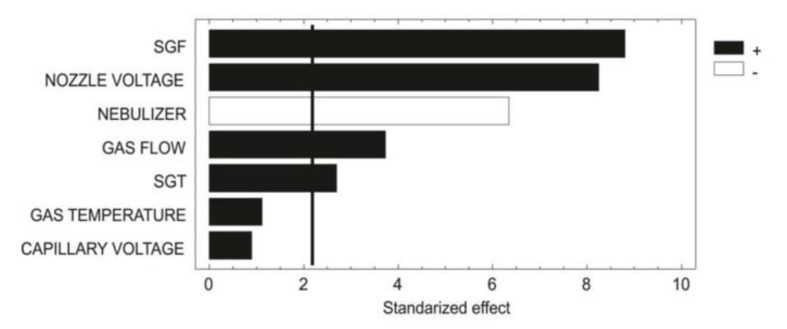
Pareto chart showing distinct factor effect patterns.

**Figure 2 molecules-25-04241-f002:**
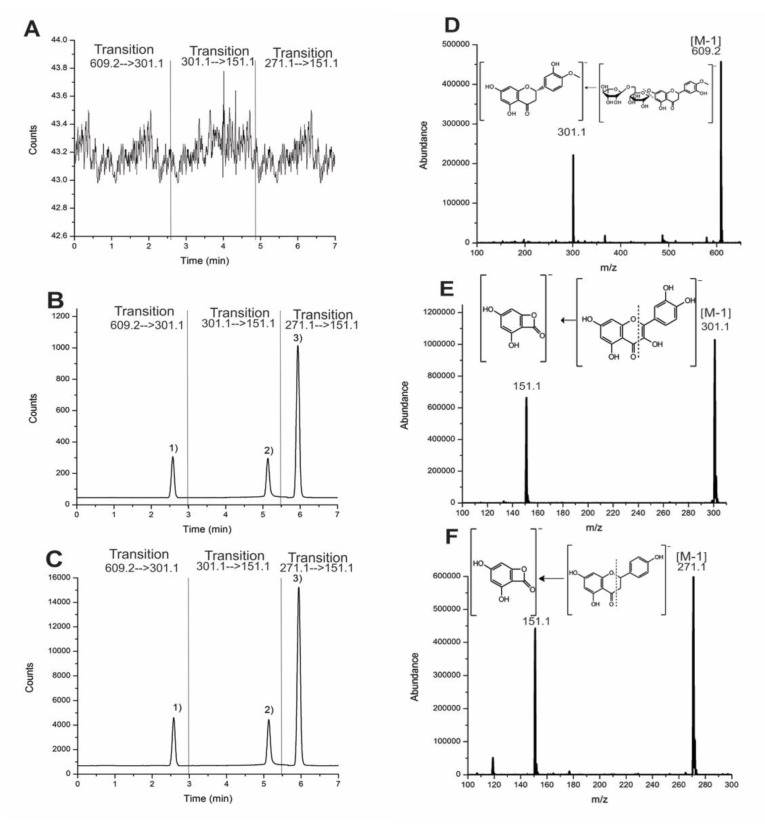
Comparison between (**A**) blank plasma chromatogram, (**B**) central point, and (**C**) optimal point. Chromatograms and mass spectra with transition at 10 ng/mL: (1) Hesperidin (**D**, 2.58 min), (2) quercetin (**E**, 5.13 min), and (3) naringenin (**F**, 5.93 min).

**Figure 3 molecules-25-04241-f003:**
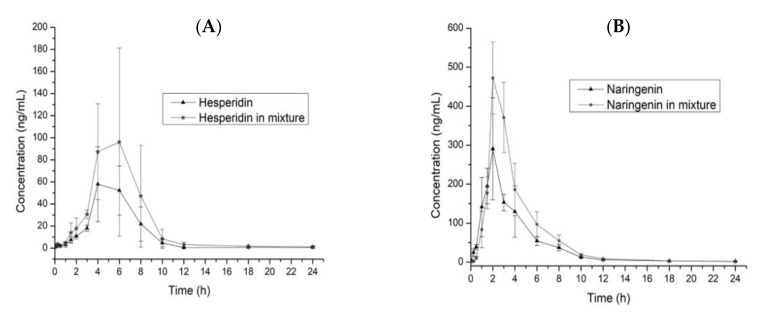
Comparison of the plasma concentration time-curves for Hsn and Hsn/Ngn (**A**) and Ngn and Ngn/Hsn (**B**). Values are expressed as the mean ± SD.

**Table 1 molecules-25-04241-t001:** Calibration parameters for hesperidin, quercetin, and naringenin.

Parameters	Hesperidin	Quercetin (IS)	Naringenin
Slope (b)	23.02	192.08	348.31
Intercept (a)	8.79	−271.68	288.7
b ± t_(__α/__2)_S_b_	23.02 ± 3.26	192.081 ± 27.34	348.31 ± 37.29
a ± t_(__α/__2)_S_a_	8.79 ± 21.64	271.68 ± 534.12	288.72 ± 416.59
r	0.9988	0.9991	0.9995
LOD (ng/mL)	0.3	0.1	0.2
LOD (pg)	0.7	0.2	0.3
LOQ (ng/mL)	1.1	0.4	0.5
LOQ (pg)	2.3	0.8	1.1

IS: Internal standard.

**Table 2 molecules-25-04241-t002:** Intra-day and inter-day precision and accuracy.

Analytes	Normal Concentration (ng/mL)	Intra-Day (*n* = 9)			Intra-Day (*n* = 27)		
		Concentration Measured (ng/mL)	Precision (RSD%)	Accuracy (%)	Concentration Measured (ng/mL)	Precision (RSD%)	Accuracy (%)
Hesperidin	2.50	2.24	4.67	89.37	2.31	9.44	92.44
	7.50	6.98	3.95	93.02	6.92	7.12	92.23
	12.50	11.63	2.71	93.04	11.64	4.86	93.12
Quercetin	2.50	2.17	6.2	86.58	2.28	5.78	91.09
	7.50	7.19	5.93	95.84	6.95	5.74	92.6
	12.50	11.99	4.05	95.86	11.77	5.56	94.16
Naringenin	2.50	2.22	3.21	88.82	2.34	8.33	93.65
	7.50	7.20	3.19	96.03	7.17	7.23	95.54
	12.50	12.05	3.09	96.34	12.17	4.82	97.32

**Table 3 molecules-25-04241-t003:** Extraction recovery and matrix effect of flavonoids (*n* = 6).

Analytes	Normal Concentration (ng/mL)	R1 Mean ± RSD	R2 Mean± RSD	R3 Mean ± RSD	Extraction Recovery (%)		Matrix Effect (%)	
					Mean	RSD	Mean	RSD
Hesperidin	2.5	266 ± 8%	268 ± 11%	250 ± 7%	93%	8%	101%	2%
	25	2472 ± 10%	2596 ± 3%	2315 ± 13%	89%	11%	105%	2%
	250	25,957 ± 6%	26,476 ± 9%	24,800 ± 9%	94%	6%	102%	5%
Quercetin	2.5	330 ± 4%	349 ± 6%	309 ± 11%	88%	6%	106%	4%
	25	3065 ± 5%	3249 ± 4%	2929 ± 6%	90%	10%	106%	2%
	250	35,704 ± 8%	36,418 ± 7%	34,747 ± 9%	95%	4%	102%	3%
Naringenin	2.5	1137 ± 5%	1160 ± 8%	1092 ± 12%	94%	9%	102%	4%
	25	10,962 ± 7%	11,291 ± 3%	10,266 ± 8%	91%	6%	103%	6%
	250	128,126 ± 3%	133,251 ± 9%	122,411 ± 5%	92%	4%	104%	1%

R1: Peak area of pure standard solution; R2: Peak area of analytes spiked post-extraction; R3: Peak area of analytes spiked before extraction. Extraction recovery (%) calculated as (R3/R2) and matrix effect (5) calculated as (R2/R1).

**Table 4 molecules-25-04241-t004:** Pharmacokinetic parameters of hesperidin, naringenin and both in a mixture by intragastric administration.

Parameter (Units)	Mean ± SD			
	Hesperidin	Hesperidin in Mixture	Naringenin	Naringenin in Mixture
T_1/2_ (h)	3.03 ± 0.71	3.14 ± 0.72	5.13 ± 2.00	4.09 ± 1.75
T_max_ (h)	4.67 ± 1.15	4.63 ± 1.08	1.67 ± 0.58	2.00
C_max_ (ng/mL)	71.89 ± 9.60	120.0 ± 72.46	310.35 ± 103.27	472.31 ± 92.19
AUC_0–24_ (ng/mL h)	288.92 ± 35.14	516.22 ± 271.70	1006.11 ± 130.46	1538.14 ± 191.47
V ((mg)/(ng/mL))	0.34 ± 0.15	0.20 ± 0.14	0.17 ± 0.07	0.09 ± 0.02
CL ((mg)/(ng/mL)/h)	0.08 ± 0.02	0.04 ± 0.02	0.02 ± 0.00	0.04 ± 0.01
MRT (h)	6.15 ± 0.79	6.51 ± 1.08	4.45 ± 0.29	4.38 ± 0.13

**Table 5 molecules-25-04241-t005:** Conditions and factors for the PBD.

Factors	Level			
	Units	−1	0	+1
Gas temperature	°C	250	300	350
Gas flow	mL/min	4	7	10
Nebulizer	psi	30	45	60
Sheath gas temperature (SGT)	°C	200	250	300
Sheath gas flow (SGF)	mL/min	6	9	12
Capillary Voltage	Volts	3000	3500	4000
Nozzle Voltage	Volts	300	500	700

**Table 6 molecules-25-04241-t006:** Factors and levels for the BBD.

Factors	Level			
	Units	−1	0	+1
Nebulizer	psi	40	50	60
SGT	°C	300	350	400
Nozzle Voltage	Volts	700	1100	1500

**Table 7 molecules-25-04241-t007:** MRM parameters for hesperidin, naringenin and internal standard.

Segment (min)	Compound	Transition (*m*/*z*)	Fragmentor (V)	Collision Energy (V)	Cell Accelerator (V)
2.76–4.0	Hesperidin	609.2 → 301.1	80	30	5
4.01–5.65	Quercetin (IS)	301.1 → 151.1	120	25	5
5.66–6.30	Naringenin	271.1 → 151.1	120	20	5

IS: Internal standard.

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
