# Peer review of "Development and Optimization of a High Sensitivity LC-MS/MS Method for the Determination of Hesperidin and Naringenin in Rat Plasma: Pharmacokinetic Approach"

_molecules, 2020, doi:10.3390/molecules25184241_

Round 1

Reviewer 1 Report

This manuscript describes a LC-MSMS method for analysis of two drugs in rat plasma.  The work is solid quantitative analysis and is properly validated.  It is applied to a pharmacokinetic study.  The work can be published but some fundamental corrections to the analytical chemistry are needed.

  1. In the introduction, there is no discussion of other LC and LCMS methods in the literature.  I see some later in results and discussion; these should be introduced up front. 
  2. In the discussion of the statistical optimization, how does the statistical optimization compare to "common sense" or classical optimization?   If the same or similar result would have been obtained why do the statistics?
  3. On line 121 and throughout, the plural of spectrum is spectra.  Please correct.
  4. In Table 1, the slope and intercept need units.  Also the LOD and LOQ values should be reported to one significant digit. In the discussion following Table 1, is it possible that the reported LOD and LOQ are different from the literature values because the other authors calculated them differently?   The calculation method can have a major impact on the results.  If the other authors did not state how calculated, you cannot make the comparison.
  5. In Table 2, pg is not a concentration unit.  Also, the measured concentrations should be reported to at most three significant digits and given how low they are and how big some of the error bars are, two significant digits is probably better.
  6. In Table 3, do not mix uncertainties and RSD.  Also uncertainties are also reported to one significant digit and the number of decimal places should match.  For example on the first line under R1, the entry should be 266 +/- 8.  The recovery and matrix effect should be reported to percent.  For example the first entry under "Extraction Recovery - Mean" should read 93% and "RSD" should read 8%.  Percent can be moved to the heading as well.
  7. In Table 4, given that the error bars seen in Figure 3 are quite large, the results should be presented with two significant digits.  

Author Response

Thank you so much for your comments, all of each were very really enriching for our paper, below we attend each of one.

Point 1: In the introduction, there is no discussion of other LC and LCMS methods in the literature.  I see some later in results and discussion; these should be introduced up front.

Response 1: We add this information in the introduction section.

Point 2: In the discussion of the statistical optimization, how does the statistical optimization compare to "common sense" or classical optimization?   If the same or similar result would have been obtained why do the statistics?

Response 2: We think that would not obtain similar results based on "common sense" and it is possible to use several resources to obtain this sensibility in the method. We think that applying a mathematical model is the best option to understand the optimization process, to choose the road to increase the sensibility in less time if doing without statistics.

Point 3: On line 121 and throughout, the plural of spectrum is spectra.  Please correct.

Response 3: Done

Point 4: In Table 1, the slope and intercept need units. Also the LOD and LOQ values should be reported to one significant digit. In the discussion following Table 1, is it possible that the reported LOD and LOQ are different from the literature values because the other authors calculated them differently?   The calculation method can have a major impact on the results.  If the other authors did not state how calculated, you cannot make the comparison.

Response 4: The injection volume to each chromatographic analysis was two microliters; for this reason, as an example, the LOD of Hesperidin was 0.34 ng/mL, which is the same as 0.34 pg/microliter, and we injected two microliters, the mass loads is 0.68 pg. We consider it essential to publish these results because, when evaluated the relationship signal-to-noise ratio, it is better in mass loads than concentration value. In section 2.2.2, we include the injection volume. We change the LOD and LOQ values to one significant digit.

Point 5: In Table 2, pg is not a concentration unit.  Also, the measured concentrations should be reported to at most three significant digits and given how low they are and how big some of the error bars are, two significant digits is probably better

Response 5: To give both reviewers a homogeneous answer, we decided to change the units in table 2 to ng/mL and reported with two significant digits to better compression and comparison with other methodologies. In table 3, all the data reported to one significant digit. 

Point 6: In Table 3, do not mix uncertainties and RSD.  Also uncertainties are also reported to one significant digit and the number of decimal places should match.  For example on the first line under R1, the entry should be 266 +/- 8.  The recovery and matrix effect should be reported to percent.  For example the first entry under "Extraction Recovery - Mean" should read 93% and "RSD" should read 8%.  Percent can be moved to the heading as well.

Response 6: We change the concentration units to ng/mL and report the results with one significant digit, and we done the suggest in point six and change the recovery and matrix effect to percent units.

Point 7: In Table 4, given that the error bars seen in Figure 3 are quite large, the results should be presented with two significant digits. 

Response 7: We change the results with two significant digits in all parameters.  

Reviewer 2 Report

I have reviewed the manuscript, ‘Development and optimization of a high sensitivity 2 LC-MS/MS method for the determination of 3 hesperidin and naringenin in rat plasma: 4 Pharmacokinetic approach’.  The paper matches the scope of the journal and is generally written in clear English, except for some grammatical errors.  It might be best to keep to consistent use of past tense for the Authors’ own work and conceptual ideas, e.g. the final paragraph in the introduction and PK section.  Overall, I found the research to be well-executed.  However, the Authors did not explain sufficiently why quercetin was included in their method, when they did not include it in their PK assessment.  I did have a few other comments or asked for clarification, as outlined below.

Line 32.  Please include references for the claim that over 60 polyphenols have been identified.

Line 33.  Change on glycosides to as glycosides.

Section 2.1.1 and 2.1.2.  This type of optimization is perhaps done intuitively by most experienced mass-spectroscopists.  However, it is an interesting demonstration that critical parameters could be optimised with this mathematical modelling approach.  When there are multiple target compounds in a method, some settings, such as needle voltage and Sheath Gas Flow, may not necessarily result in the same optimised response for each compound.  Can the authors provide some comment on whether both compounds independently gave the same results?  My thinking is that equation 1 (line 100) must surely relate to one of the target compounds only (BBD) as it is based on standard curve values.  Could the other compound perform better if it was optimised in the same way?  If so, then a compromise would be necessary, since the instrument settings (gas flows & temp) cannot be changed during a run on most spectrometers.

Line 121.  The plural of spectrum is spectra.

Line 136, Table 1.  Concentrations and mass loads are reported in the Table and elsewhere in the manuscript.  However, the Authors did not include the injection volumes for it to be clear how the on-column mass loads were obtained.

Lines 143-152.  Please be consistent with the use of units.  It is difficult to follow the comparisons when the Authors switch between concentrations and mass loads.

Line 153.  What is the relevance of estrogens in the context of the manuscript?

Line 169 Table 2 and 189 Table 3.  Please change the nominal concentration column headings to on-column mass loads.  The Authors are not using concentration units.  I also suspect that the accuracy is overstated by the use of 2 decimal places for the reporting of accuracy and precision.

Line 177-180.  I think this statement is redundant.  Any person developing a LC-ESI/MS method knows that these parameters require optimisation.

Line 201 Table 4.  The methods section states that the calibration curves ranged from 5-25 ng/mL.  In the Table, Cmax values well above that range are reported.  Please provide the matching validated calibration curve data in the methods and results. 

Line 207.  I could not find any information in Table 4 which suggested a 1.04 h reduction in elimination rate.  Please clarify.

Line 223.  The Authors state that Cmax did not increase for the mixture.  In fact, it almost doubled for both compounds (Table 4).

Line 230-236.  This paragraph was difficult to follow with the extent of information provided to the readership.  It is only apparent from the mass spectra that one compound was used as its glycoside and the other the aglycone.  I would recommend that the difference is made clear at the start of the paragraph and also earlier in the document.  The question also arises, why did the Authors choose to use these different forms of the compounds.  Surely the PK would be dramatically affected, as confirmed by the results in Table 4.  Please provide some rationale.

Line 251.  Was ethics approval sought/granted for the study?  If so, please state the approval number or if not, why was ethics approval not deemed necessary for the animal study.

Line 297.  Weights in the pg range cannot be measured.  Please provide the actual procedure for making up the solutions to give the desired target loads in 100 microlitre of plasma.

Line 326.  Please state the injection volume.

Line 332.  The sentence does not make sense (…continue stability in such values…).  Please rephrase.

Line 334.  The Authors cannot claim that the entire analysis took 7 min.  They chose to detect over this interval, but in fact the overall method was 22 min overall with column clean-up and equilibration time added.

Line 341.  Please state the volume of blood taken each time.

Line 356.  Once again, the Authos state that Cmax did not vary significantly.  That contradicts the information provided in Table 4.

Line 359.  The Authors referred to a volume of 100 microlitres earlier in the manuscript as though that that the volume required for determining the plasma concentrations.  A total of 17 samples were taken (methods, line 324/3), so that should bring the total volume to 1.7 mL.  Please clarify.

Author Response

Thank you so much for your comments, all of each were very really enriching for our paper, below we attend each of one.

Point 1: The Authors did not explain sufficiently why quercetin was included in their method, when they did not include it in their PK assessment. 

Response 1: We use quercetin as an internal standard, the aim of the work was developing a high sensitivity LC-MS method to applied in pharmacokinetic studies from hesperidin and naringenin. In this case, to have high-quality control, we consider it very important to include an internal standard, all calibrations were based on this methodology.

Point 2: Line 32.  Please include references for the claim that over 60 polyphenols have been identified.

Response 2: We include the reference of Tripoli and coworkers [1] and change the sentence to: “according with Tripoli and coworkers [1] cover over 60 types have been identified”

  1. Tripoli, E.; La Guardia, M.; Giammanco, S., Di Majo, D.; Giammanco, M. Citrus flavonoids: Molecular structure, biological activity and nutritional properties: A review. Food Chem. 2007, 104, 466-479. https://doi.org/10.1016/j.foodchem.2006.11.054

Point 3: Line 33.  Change on glycosides to as glycosides.

Response 3: Done

Point 4: Section 2.1.1 and 2.1.2.  This type of optimization is perhaps done intuitively by most experienced mass-spectroscopists.  However, it is an interesting demonstration that critical parameters could be optimised with this mathematical modelling approach.  When there are multiple target compounds in a method, some settings, such as needle voltage and Sheath Gas Flow, may not necessarily result in the same optimised response for each compound.  Can the authors provide some comment on whether both compounds independently gave the same results?  My thinking is that equation 1 (line 100) must surely relate to one of the target compounds only (BBD) as it is based on standard curve values.  Could the other compound perform better if it was optimised in the same way?  If so, then a compromise would be necessary, since the instrument settings (gas flows & temp) cannot be changed during a run on most spectrometers.

Response 4: When we evaluate a PBD and BBD, we observed a similar result with hesperidin, quercetin, and naringenin; as reviewer two mentioned, it difficult to change the spectrometers settings during the chromatographic analysis. We attributed these similitudes with the negative ionization of flavonoids, due to the poli-hydroxyl groups. The optimization process allows us to raise each flavonoid's sensitivity to 15 times concerning the central points. This method was tested with six flavonoids like rutin, hesperidin, naringin, quercetin, naringenin, and hesperetin, and similar results were observed. We attach a chromatogram with comparative the central point (gray) and optimal conditions (blue).

Point 5: Line 121.  The plural of spectrum is spectra.

Response 5: Done

Point 6: Line 136, Table 1.  Concentrations and mass loads are reported in the Table and elsewhere in the manuscript.  However, the Authors did not include the injection volumes for it to be clear how the on-column mass loads were obtained.

Response 6: The injection volume to each chromatographic analysis was two microliters; for this reason, as an example, the LOD of Hesperidin was 0.34 ng/mL, which is the same as 0.34 pg/microliter, and we injected two microliters, the mass loads is 0.68 pg. We consider it essential to publish these results because, when evaluated the relationship signal-to-noise ratio, it is better in mass loads than concentration value. In section 2.2.2, we include the injection volume.

Point 7: Lines 143-152.  Please be consistent with the use of units.  It is difficult to follow the comparisons when the Authors switch between concentrations and mass loads.

Response 7: We change the units to ng/mL to be consistent and homogenized to understand the comparisons.

Point 8: Line 153.  What is the relevance of estrogens in the context of the manuscript?

Response 8: The sentence about estrogens refers only to highly sensitive pharmacokinetics; it is an introduction to knowns that report other methodologies that use a few blood microliters and, it is possible to study pharmacokinetics parameters by UHPLC-MS/MS.

Point 9: Line 169 Table 2 and 189 Table 3.  Please change the nominal concentration column headings to on-column mass loads.  The Authors are not using concentration units.  I also suspect that the accuracy is overstated by the use of 2 decimal places for the reporting of accuracy and precision.

Response 9: To give both reviewers a homogeneous answer, we decided to change the units in table 2 to ng/mL and reported with two significant digits to better compression and comparison with other methodologies. In table 3, all the data reported to one significant digit. 

Point 10: Line 177-180.  I think this statement is redundant.  Any person developing a LC-ESI/MS method knows that these parameters require optimisation.

Response 10: We eliminate the statement.

Point 11: Line 201 Table 4.  The methods section states that the calibration curves ranged from 5-25 ng/mL.  In the Table, Cmax values well above that range are reported.  Please provide the matching validated calibration curve data in the methods and results.

Response 11: Thanks for the comment, we have a mistake, the linearity range for each flavonoid was 5 to 1,000 ng/mL. We change this information in 3.4.1 section.

Point 12: Line 207.  I could not find any information in Table 4 which suggested a 1.04 h reduction in elimination rate.  Please clarify

Response 12: When naringenin administrating as pure flavonoid, the T1/2 was 5.13 h, and, when the administration of naringenin is in mixture with hesperidin, the T1/2 reduces 1.04 h, in table 4 show the results like 4.09 h. The subtraction of these values (5.13-4.09) showed a difference of 1.04 h. It is necessary to mention this information, due to this parameter does not show statistical differences.

Point 13: Line 223.  The Authors state that Cmax did not increase for the mixture.  In fact, it almost doubled for both compounds (Table 4).

Response 13: We change the statement, to "the Cmax did not show a significant difference (p>0.05), but the concentration of hesperidin increases around 1.6 times when administered in the mixture, the same case was observed with naringenin", this explains that even though do not exist statistical differences, the Cmax increase around 1.6 times.

Point 14: Line 230-236.  This paragraph was difficult to follow with the extent of information provided to the readership.  It is only apparent from the mass spectra that one compound was used as its glycoside and the other the aglycone.  I would recommend that the difference is made clear at the start of the paragraph and also earlier in the document.  The question also arises, why did the Authors choose to use these different forms of the compounds.  Surely the PK would be dramatically affected, as confirmed by the results in Table 4.  Please provide some rationale.

Response 14: We chose to study hesperidin and naringenin because in our research group we isolated these flavonoids from different citrus fruits from the Mexican region, an in reference 8 (Sánchez-Recillas, 2019) a paper our working group showed that the mixture of these flavonoids has a different pharmacological effect, like vasorelaxant, antihypertensive and antihyperglycemic. This molar mixture of hesperidin and naringenin is found in citrus. For this reason, we study the pharmacokinetic and development of the analytical method.

Point 15: Line 251.  Was ethics approval sought/granted for the study?  If so, please state the approval number or if not, why was ethics approval not deemed necessary for the animal study.

Response 15: In section 3.2, we demonstrated have the experimental conditions according with Mexican regulations and international guidelines.

Point 16: Line 297.  Weights in the pg range cannot be measured.  Please provide the actual procedure for making up the solutions to give the desired target loads in 100 microlitre of plasma.

Response 16: We change to concentration units in ng/mL

Point 17: Line 326.  Please state the injection volume.

Response 17: Done, the injection volume was 2 microliters

Point 18: Line 332.  The sentence does not make sense (…continue stability in such values…).  Please rephrase.

Response 18: We delete "to continue stability in such values" to clarify the sentence.

microliters

Point 19: Line 334.  The Authors cannot claim that the entire analysis took 7 min.  They chose to detect over this interval, but in fact the overall method was 22 min overall with column clean-up and equilibration time added.

Response 19: We delete "The entire analysis of the sample took 7 min"

Point 20: Line 341.  Please state the volume of blood taken each time.

Response 20: Done, we have taken 0.5 mL of total blood.

Point 21: Line 356.  Once again, the Authos state that Cmax did not vary significantly.  That contradicts the information provided in Table 4.

Response 21: The T-Student analysis does not have a statistical difference in Cmax because the standard deviation is high. However, we clarify that even though there is no statistical difference in Cmax, this increases around 1.6 times when it is administrated in a mixture of both flavonoids. We add this information in conclusions.

Point 22: Line 359.  The Authors referred to a volume of 100 microlitres earlier in the manuscript as though that that the volume required for determining the plasma concentrations.  A total of 17 samples were taken (methods, line 324/3), so that should bring the total volume to 1.7 mL.  Please clarify.

Response 22: We need only 100 microlitres of sample to determine the plasma concentrations of both flavonoids. We have taken 17 samples in total, one for each time of analysis. We add this information in conclusions.

Round 2

Reviewer 1 Report

Authors have adequately responded to criticisms.  

Reviewer 2 Report

I thank the Authors for addressing all my comments and queries.  i am satisfied that the manuscript is scientifically sound.  However, some editing for English is recommended.  E.g., in the revised version, line 32, the inserted phrase is very poor English.

I would also suggest that the half-lives and Cmax shown in Table 4 are revised to give fewer decimal places.  The accuracy of the measurements is overstated.